# Prognostic Modeling of Overall Survival in Glioblastoma Using Radiomic Features Derived from Intraoperative Ultrasound: A Multi-Institutional Study

**DOI:** 10.3390/cancers17020280

**Published:** 2025-01-16

**Authors:** Santiago Cepeda, Olga Esteban-Sinovas, Vikas Singh, Aliasgar Moiyadi, Ilyess Zemmoura, Massimiliano Del Bene, Arianna Barbotti, Francesco DiMeco, Timothy Richard West, Brian Vala Nahed, Giuseppe Roberto Giammalva, Ignacio Arrese, Rosario Sarabia

**Affiliations:** 1Department of Neurosurgery, Río Hortega University Hospital, 47014 Valladolid, Spain; oestebans@saludcastillayleon.es (O.E.-S.); iarreser@saludcastillayleon.es (I.A.); rsarabia@saludcastillayleon.es (R.S.); 2Department of Neurosurgery, Tata Memorial Hospital, TMC and Homi Bhabha National Institute, Mumbai 400012, Maharashtra, India; drvikaskumarsingh@gmail.com (V.S.); aliasgar.moiyadi@gmail.com (A.M.); 3UMR 1253, iBrain, Université de Tours, Inserm, 37000 Tours, France; ilyess.zemmoura@univ-tours.fr; 4Department of Neurosurgery, CHRU de Tours, 37000 Tours, France; 5Department of Neurosurgery, Fondazione IRCCS Istituto Neurologico Carlo Besta, Via Celoria 11, 20133 Milan, Italy; macs.delbene@gmail.com (M.D.B.); ariannabarbotti.ab@gmail.com (A.B.); francesco.dimeco@istituto-besta.it (F.D.); 6Department of Pharmacological and Biomolecular Sciences, University of Milan, 20122 Milan, Italy; 7Department of Oncology and Hematology-Oncology, Università Degli Studi di Milano, 20122 Milan, Italy; 8Department of Neurological Surgery, Johns Hopkins Medical School, Baltimore, MD 21205, USA; 9Department of Neurosurgery, Massachusetts General Hospital, Mass General Brigham, Harvard Medical School, Boston, MA 02114, USA; trwest@mgh.harvard.edu (T.R.W.); bnahed@mgh.harvard.edu (B.V.N.); 10Department of Neurosurgery, ARNAS Civico Di Cristina Benfratelli Hospital, 90127 Palermo, Italy; robertogiammalva@live.it

**Keywords:** intraoperative ultrasound, glioblastoma, radiomics, survival

## Abstract

This study evaluates the potential utility of intraoperative ultrasound (iUS) radiomics for prognostic modeling in glioblastoma patients. Using data from a multicenter cohort, models combining radiomic features with clinical variables were compared to those based on individual data types. The results suggest that integrating iUS radiomics and clinical data may improve survival prediction. These findings highlight the promise of iUS as a complementary tool for prognostic assessment in glioblastoma, warranting further investigation.

## 1. Introduction

Glioblastoma is the most aggressive and prevalent primary brain tumor, accounting for approximately 80% of all malignant brain neoplasms [1]. Despite advances in multimodal therapies, the median overall survival (OS) remains dismal at approximately 15 months [2,3]. This reality underscores the critical need to develop robust and reliable prognostic tools to guide personalized treatment strategies. Accurate survival prediction is essential for optimizing therapeutic decisions, such as determining the extent of surgical resection and tailoring adjuvant therapies to improve patient outcomes.

Radiomics, a rapidly advancing field at the intersection of medical imaging and artificial intelligence (AI), has emerged as a transformative approach in oncology for developing prognostic models [4,5]. By extracting high-dimensional quantitative features from imaging modalities, including magnetic resonance imaging (MRI), positron emission tomography (PET), and computed tomography (CT), radiomics enables a deeper understanding of tumor biology [6]. These features, often imperceptible to the human eye, quantify variations in intensity, shape, texture, and heterogeneity, offering valuable insights into the underlying physical and biological properties of tumors [7,8]. Extensive research has applied radiomic analyses to MRI and PET in glioblastoma, demonstrating their potential to predict survival and treatment response [9,10,11,12,13,14,15].

Intraoperative ultrasound (iUS) is a widely used real-time imaging modality for tumor localization and resection guidance during surgery [16,17,18,19]. Unlike MRI or PET, iUS operates based on the acoustic properties of tissues, such as sound wave reflection, attenuation, and elasticity [20]. These characteristics provide a unique perspective on the tumor microenvironment, potentially revealing complementary biological and physical information.

This study builds on the hypothesis that radiomic features derived from iUS capture a distinct set of physical and biological properties compared with those obtained from other image modalities [21]. Preliminary work by our group has demonstrated the feasibility of extracting radiomic features from iUS in brain tumors, whereas similar methodologies have been successfully applied by other researchers in different neoplasms [22,23,24,25]. The innovation of this study lies in the application of iUS-based radiomics for prognostic modeling in glioblastoma patients via a multi-institutional cohort. By leveraging the unique properties of iUS, this study aims to explore its potential as a prognostic tool and contribute novel insights into personalized management strategies in glioblastoma.

## 2. Materials and Methods

### 2.1. Study Population

The primary dataset for this study was obtained from the BraTioUS Consortium (ClinicalTrials.gov Identifier: NCT05062772), a multicenter collaboration including six institutions: Río Hortega University Hospital (RHUH), Valladolid, Spain; Tata Memorial Center (TMC), Mumbai, India; Istituto Neurologico “Carlo Besta” (INCC), Milan, Italy; University of Palermo (UPALER), Palermo, Italy; Le Centre Hospitalier Régional Universitaire de Tours (CHRUT), Tours, France; and Massachusetts General Hospital (MGH), Boston, MA, USA. This dataset comprises intraoperative ultrasound (iUS) images from patients who underwent brain tumor surgery between 2018 and 2023.

For this study, we included patients with a confirmed diagnosis of glioblastoma, IDH wild type, and grade 4, based on the 2021 WHO Classification of Central Nervous System Tumors [26], who underwent surgery followed by the Stupp protocol (maximal safe resection, concurrent radiotherapy, and temozolomide, followed by adjuvant temozolomide cycles) [2]. Only preresection B-mode iUS images were analyzed. The primary endpoint was overall survival (OS), defined as the number of days from the initial pathological diagnosis to death (censored = 1) or the last recorded date the patient was known to be alive (censored = 0). All included patients either had a minimum follow-up of one year or had reached the study endpoint prior to this timeframe. Additional clinical variables, including age, preoperative Karnofsky performance status (KPS), extent of resection (EOR), and initial tumor volume, were also collected.

Patients were excluded if they had alternative histopathological diagnoses, suboptimal iUS image quality or artifacts that hindered analysis, missing clinical data, or insufficient follow-up. The use of anonymized data was approved by the Research Ethics Committee (CEIm) at Río Hortega University Hospital, Valladolid, Spain (Approval number 21-PI085. Date: 30 April 2021).

### 2.2. Ground Truth Segmentation

For each patient, one 2D iUS slice showing the largest tumor diameter was selected for analysis. The tumors in the selected slices were manually segmented via ITK-SNAP software (version 4.0.1, http://itksnap.org, accessed on 15 June 2024), ensuring the exclusion of large necrotic or cystic regions. All segmentations were performed by an experienced neurosurgeon with 12 years of expertise in medical imaging, specifically in the interpretation and analysis of intraoperative ultrasound.

### 2.3. Image Preprocessing and Radiomic Feature Extraction

The iUS images, which were originally stored in portable network graphics (PNG) format, were converted to 2D NIfTI (Neuroimaging Informatics Technology Initiative) format to ensure compatibility with the radiomic feature extraction tools. Following the conversion, intensity normalization was performed via z-score normalization, standardizing the intensity values for each image to have a mean of zero and a standard deviation of one. Tumor regions of interest (ROIs) were defined via ground truth segmentations obtained from manual annotations, which were used as masks for the extraction of radiomic features.

Radiomic features were extracted via the PyRadiomics library, version 3.1.0 [27], adhering to the IBSI (Image Biomarker Standardization Initiative) guidelines [28] to ensure reproducibility and consistency, acknowledging that some minor differences exist as documented by the PyRadiomics development team. The extraction process included features from three image types: Original, Laplacian of Gaussian (LoG), and Wavelet. The LoG filter, with sigma values of 2.0, 3.0, 4.0, and 5.0, was used to enhance texture patterns at multiple spatial scales. The wavelet filter decomposes images into frequency subbands, capturing detailed information on image intensity variations.

The extracted features were grouped into several classes, including shape2D, first-order statistics, and texture-based features derived from GLCM (gray-level co-occurrence matrix), GLRLM (gray-level run length matrix), GLSZM (gray-level size zone matrix), GLDM (gray-level dependence matrix), and NGTDM (neighboring gray-tone difference matrix). These features provide a comprehensive quantitative description of tumor morphology, intensity distribution, and texture heterogeneity.

To ensure comparability across images, the preprocessing pipeline included resampling to a uniform pixel spacing of [2, 2] mm using B-spline interpolation. A padding distance of 10 pixels was applied around the ROI to fully capture the tumor region, and the images were cropped accordingly.

### 2.4. Statistical Analysis

The primary endpoint was OS, defined as the number of days from the initial pathological diagnosis to death (censored = 1) or the last known date the patient was alive (censored = 0). To ensure robust performance evaluation and generalizability, the dataset was randomly divided into five folds for cross-validation, stratified by center.

The clinical variables included age, KPS, EOR (categorized as complete or incomplete), and initial tumor volume. Instances containing missing values (NaNs) were excluded from the dataset to ensure the integrity of the analysis. Radiomic variables were then standardized by centering (subtracting the mean) and scaling (dividing by the standard deviation) using statistics calculated exclusively from the training dataset. This approach was applied to prevent information leakage from the training set into the testing set, thereby ensuring a fair evaluation of the predictive models. Radiomic feature selection was performed on the training sets via the minimum redundancy maximum relevance (mRMR) [29] method to minimize feature redundancy and retain the most informative predictors. A maximum of 20 radiomic features were selected based on the mRMR results to construct the radiomic models.

For each fold, three Cox proportional hazards models were developed: (1) a radiomic model based on the features selected by mRMR; (2) a clinical model incorporating age, KPS, EOR, and initial tumor volume; and (3) a combined model integrating both clinical and radiomic features. Model fitting was performed using stepwise selection guided by the Akaike information criterion (AIC) to identify the optimal set of predictors.

The performance of the models was assessed via the concordance index (C-index) [30], a metric that quantifies the model’s ability to discriminate survival outcomes. The C-index was calculated separately for the training and test datasets within each fold, enabling a comparative analysis of the performance of radiomic, clinical, and combined models. Additionally, Kaplan–Meier survival curves were generated for the test datasets within each fold to evaluate and visualize the stratification of patients into risk groups based on the models’ predictions, and the differences between survival curves were assessed using the log-rank test.

To evaluate the comparability of the training and testing datasets across folds, statistical analyses were conducted to assess the distributions of key clinical and imaging variables. For continuous variables, such as OS, age, preoperative KPS, and preoperative tumor volume, the Mann-Whitney U test was used to compare median values and interquartile ranges (IQRs) between the training and testing groups. For categorical variables, EOR, the chi-squared test was used to assess proportional differences. Statistical significance was defined as a *p*-value < 0.05.

The statistical analyses were implemented in R version 4.4.1, leveraging libraries such as survminer (version 0.4.9), mRMRe (version 2.1.2.1), caret (version 6.0.94), and glmnet (version 4.1.8) libraries. Summary statistics, including the C-index values for both the training and test datasets, were computed and averaged across the folds to ensure reliable evaluation. Figure 1 provides a schematic overview of the workflow followed in this study.

## 3. Results

The study included 114 patients diagnosed with glioblastoma from four centers. Patients from the remaining two centers in the BraTioUS dataset (UPALER and MGH) were excluded due to incomplete clinical data. The median OS was 382 days, with an IQR of 444.25 days. The mean age of the cohort was 56.9 years (SD: 13.7), and the median preoperative KPS was 80 (IQR: 20). The median preoperative tumor volume was 32.69 cm^3^ (IQR: 31.77 cm^3^). The EOR was categorized as complete in 59 patients (51.8%) and incomplete in 55 patients (48.2%). The distribution of patients and their respective clinical characteristics across centers is summarized in Table 1.

The distributions of the clinical and imaging variables between the training and testing groups across the five folds were consistent and did not significantly differ. The details are shown in Table 2.

The radiomic model exhibited moderate performance, with training C-index values ranging from 0.64 to 0.70 across folds and testing C-index values ranging from 0.60 to 0.79. Likelihood ratio tests and Wald tests were significant for the training datasets (*p* < 0.01), whereas the testing datasets yielded mixed results, with some nonsignificant *p*-values, particularly for folds 3 and 5.

The clinical model consistently outperformed the radiomic model, achieving higher training C-index values (0.68–0.72) and testing C-index values (0.66–0.82). Both the likelihood ratio and Wald tests were significant across training datasets (*p* < 0.01), whereas testing datasets yielded varying results, with significant outcomes in some folds.

The combined model demonstrated the best predictive performance among all the approaches, with training C-index values ranging from 0.76 to 0.79 and testing C-index values ranging from 0.77 to 0.91. The likelihood ratio and Wald tests were consistently significant in the training datasets (*p* < 0.01), whereas the testing datasets yielded significant results in most folds, except for some nonsignificant Wald tests in folds 3 and 5. The observed differences in C-index values between the training and validation sets likely reflect the diverse complexity of the training data, demonstrating the model’s ability to generalize effectively to unseen subsets while capturing broader patterns during training. The details are presented in Table 3. The results of the Kaplan–Meier survival curve analysis for the combined model on the test groups across the different folds are presented in Figure 2.

The analysis of radiomic features selected for the predictive models across folds revealed a diverse set of variables with varying levels of significance and contributions to the models. Eight radiomic features were consistently identified in multiple folds, with their coefficients, predominant directional effects, and statistical significance summarized in Table 4. Figure 3 illustrates the coefficients of the top feature importances for the combined model on the test datasets across folds.

Among the features, the GLRLM Run Variance from filter LoG sigma 2.0, a measure of the variability of run lengths in gray-level patterns, emerged as a significant predictor (*p* < 0.05) in three out of five folds, demonstrating a consistent positive association with outcomes (average coefficient = 0.52). Similarly, the maximal correlation coefficient (MCC) from the GLCM in filter LoG sigma 2.0, which quantifies the complexity of spatial gray-level dependencies, showed a significant negative threefold association (average coefficient = −0.32).

Features such as the median intensity from the wavelet (H) filter, long run high gray level emphasis from the wavelet (H), indicating the presence of long, high-intensity runs in the image, and GLSZM small area emphasis from LoG sigma 3.0, a measure of the concentration of small bright regions in the filtered image, exhibited partial significance (0.05 < *p* < 0.1), suggesting mixed or context-dependent contributions to the models. The directional effects for these features were predominantly positive or mixed.

Other variables, including NGTDM Busyness from LoG sigma 4.0, which measures the spatial irregularity of gray levels; NGTDM Strength from LoG sigma 4.0, which quantifies the degree of contrast enhancement due to spatial correlation; and NGTDM Busyness from the Original Image, were frequently selected but did not reach statistical significance (*p* ≥ 0.05). These features presented consistent directional effects, either positive or negative, across folds. Figure 4 illustrates representative examples of two patients, showcasing differences in overall survival and the corresponding radiomic feature maps.

## 4. Discussion

This study investigated the prognostic value of radiomic features extracted from iUS in glioblastoma and compared the performance of radiomic, clinical, and combined models in predicting OS. Our findings underscore the potential of iUS radiomics as a complementary prognostic tool while emphasizing the superior performance of combined models that integrate clinical and radiomic features.

A notable strength of this study is the use of a multi-institutional dataset, which enhances the generalizability of the findings. The cross-validation approach, stratified by centers, ensures robust performance evaluation by balancing the representation of data from multiple institutions, while standardized radiomic feature extraction minimizes bias and enhances reproducibility.

Ultrasound-based radiomics has shown promise in survival modeling across various cancer types. In endometrial cancer, Huang et al. [23] developed a combined radiomic–clinical model that outperformed standalone approaches, achieving high predictive accuracy for disease-free survival (AUC: 0.893 in training, 0.885 in validation cohorts).

Similarly, Xiong et al. [24] constructed a radiomics signature from preoperative ultrasound images in invasive breast cancer, demonstrating improved predictive performance when combined with clinicopathological data (C-index: 0.796). Yu et al. [25], in a multi-institutional study of triple-negative breast cancer, further validated the prognostic value of ultrasound radiomics, showing superior performance compared to TNM staging alone (C-index: 0.75 in the training cohort).

In glioblastoma, our study group has previously conducted the first feasibility study using iUS radiomics for OS prediction [22]. This single-center study, with a limited sample size, identified key features correlated with survival outcomes, highlighting the potential of iUS radiomics to capture tumor heterogeneity and provide prognostic insights. The primary distinctions between the previous study and the current one include the utilization of a multicenter cohort, which incorporates data from various scanner manufacturers with differing acquisition parameters. Additionally, this study employs a different radiomics feature extraction software compared to the earlier work. Building on our prior research, the current study further advances the application of iUS radiomics in glioblastoma by integrating both radiomic and clinical features into comprehensive predictive models. The combined models consistently achieved the highest concordance indices (C-index) across all folds, with a mean C-index of 0.78 (95% CI: 0.72–0.84) in the training datasets and 0.87 (95% CI: 0.76–0.98) in the testing datasets. These results outperformed the radiomic models, which achieved a C-index of 0.67 (95% CI: 0.59–0.75) in training and 0.72 (95% CI: 0.57–0.86) in testing, as did the clinical models, which achieved a C-index of 0.70 (95% CI: 0.63–0.77) in training and 0.73 (95% CI: 0.60–0.87) in testing. Statistical tests, including likelihood ratio, Wald, and log-rank tests, further validated the superiority of the combined models, with consistently higher test statistics and significance levels than standalone radiomic or clinical models. These results highlight the complementary nature of radiomic and clinical features in capturing diverse aspects of tumor biology and patient outcomes.

The radiomic features analyzed in this study provide quantifiable insights into the complex and heterogeneous biology of glioblastoma. GLRLM Run Variance (Filter: LoG, sigma = 2.0 mm), which measures the variability in the lengths of consecutive runs of similar gray-level intensity within the tumor, reflects spatial heterogeneity. A higher Run Variance indicates a more heterogeneous tumor texture, often associated with regions of mixed necrosis, cellular proliferation, and angiogenesis, which are key characteristics of glioblastoma’s aggressive biology. Similarly, the GLCM MCC (Filter: LoG, sigma = 2.0 mm) quantifies the complexity of spatial dependencies between gray-level intensities, capturing the organization or disorganization of the tumor microenvironment. A negative association with survival suggests that greater disorganization, indicative of a chaotic tissue architecture, is correlated with a worse prognosis, reflecting the invasiveness and adaptability of the tumor.

Partially significant features also offer biological insights. Median Intensity (Filter: Wavelet, H), which represents the central tendency of voxel intensities, may capture global tumor density and be related to stromal composition or cellular packing. GLRLM Long Run High Gray Level Emphasis (Filter: Wavelet, H) reflects the prevalence of elongated, high-intensity regions, potentially corresponding to well-vascularized areas or organized stromal components that contribute to tumor growth and survival. The GLSZM Small Area Emphasis (Filter: LoG, sigma = 3.0 mm) highlights the prominence of small, dense clusters within the tumor, which may represent isolated niches of active growth or necrotic cores, indicating localized aggressiveness.

Other features, such as NGTDM Busyness (filter: LoG, sigma = 4.0 mm) and NGTDM Strength (filter: LoG, sigma = 4.0 mm), capture textural irregularities and contrast-enhanced regions within the tumor. Busyness quantifies abrupt intensity transitions, potentially marking boundaries between necrotic and viable tumor areas or between the tumor and surrounding tissues. Strength reflects the degree of spatial contrast correlation, which is often linked to enhanced vascularity or dense cellular clusters. Although these features were not consistently significant, their frequent selection suggests that they may capture biologically meaningful patterns under specific imaging or tumor contexts.

Collectively, these radiomic features quantify critical aspects of glioblastoma heterogeneity, including spatial organization, density, vascularization, and textural complexity. These quantifications align with known pathological traits such as necrosis, angiogenesis, and cellular heterogeneity, providing a mechanistic basis for their prognostic value. Understanding these associations helps bridge the gap between imaging data and underlying tumor biology, emphasizing the potential of radiomics to offer nuanced, noninvasive insights into glioblastoma behavior.

Several limitations should be acknowledged in this study. The relatively small sample size, particularly in certain centers, may have restricted the statistical power to detect subtle differences in prognostic performance. Another notable limitation is the use of a single ultrasound image per patient, which, while representative, does not fully capture intratumoral heterogeneity and may omit critical information. Finally, although the models underwent rigorous internal validation, external validation with independent datasets is necessary to ensure the generalizability and robustness of these findings.

Future research should prioritize expanding the sample size to improve the statistical power and generalizability of findings, as well as increasing the number of slices analyzed per patient or transitioning to the use of full 3D volumetric data. These approaches would allow for a more comprehensive characterization of intratumoral heterogeneity, which is critical for enhancing the predictive performance of radiomic models. Furthermore, exploring and implementing advanced harmonization techniques to standardize radiomic features across datasets from different institutions or imaging protocols will be essential for ensuring model reproducibility and robustness in diverse clinical settings.

Another key avenue for investigation involves the integration of iUS data with complementary imaging modalities, such as MRI and molecular biology data, to create multimodal predictive models that leverage the strengths of each technique. One potential approach could involve collecting data from ultrasound fused with preoperative MRI, allowing co-registered images to extract features from both modalities within the same tumor region. Such integrative approaches could provide a more holistic understanding of tumor behavior and improve the accuracy of prognostic assessments.

## 5. Conclusions

This study demonstrated that iUS radiomics offers valuable prognostic insights into glioblastoma. The best performance was observed when radiomic features were combined with clinical data, highlighting the complementary nature of these information sources. Validation in larger, more diverse cohorts is essential to establish its clinical application.

## Figures and Tables

**Figure 1 cancers-17-00280-f001:**
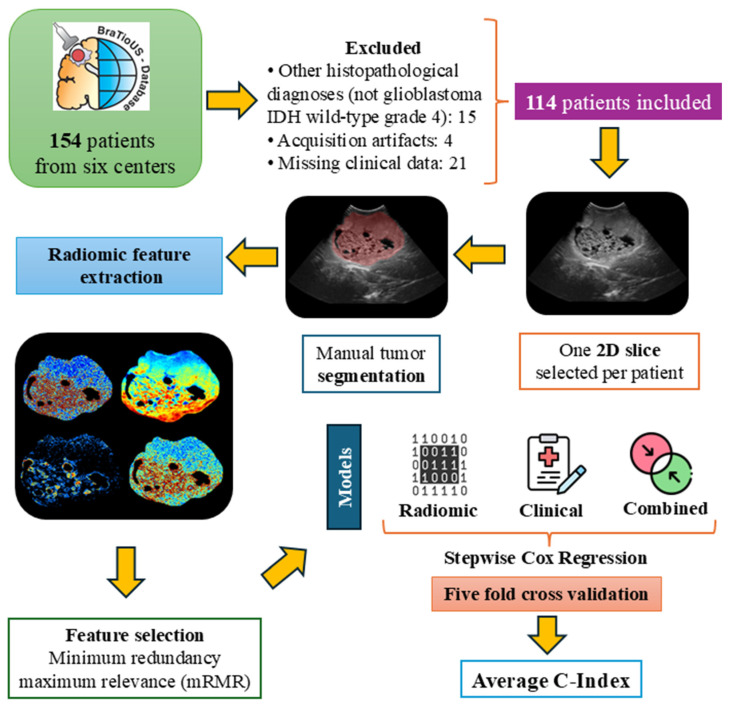
Schematic overview of the workflow used in this study.

**Figure 2 cancers-17-00280-f002:**
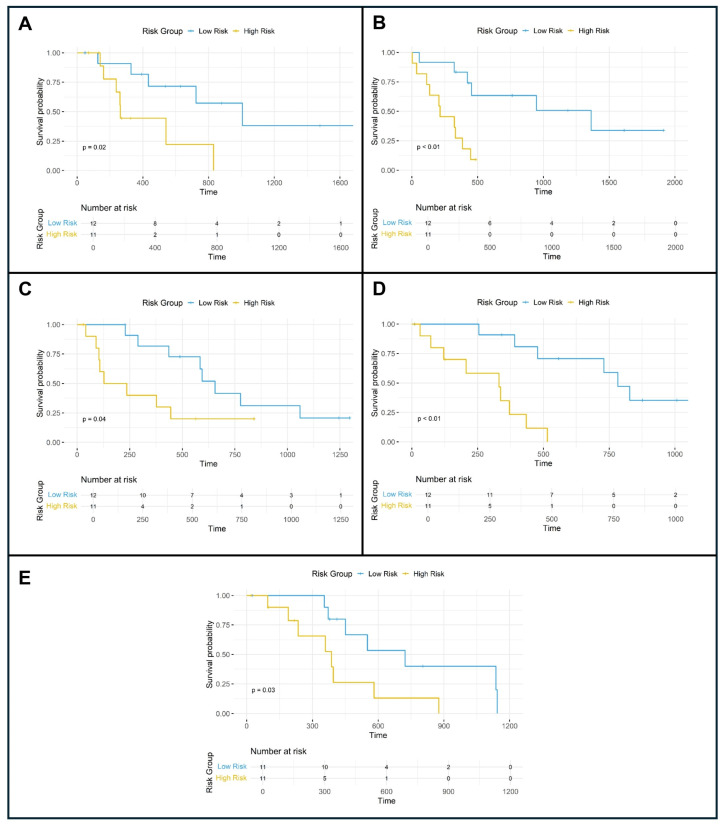
Kaplan–Meier survival curve analysis of the risk groups predicted by the combined model for the test datasets across folds 1 to 5, shown in panels (**A**–**E**).

**Figure 3 cancers-17-00280-f003:**
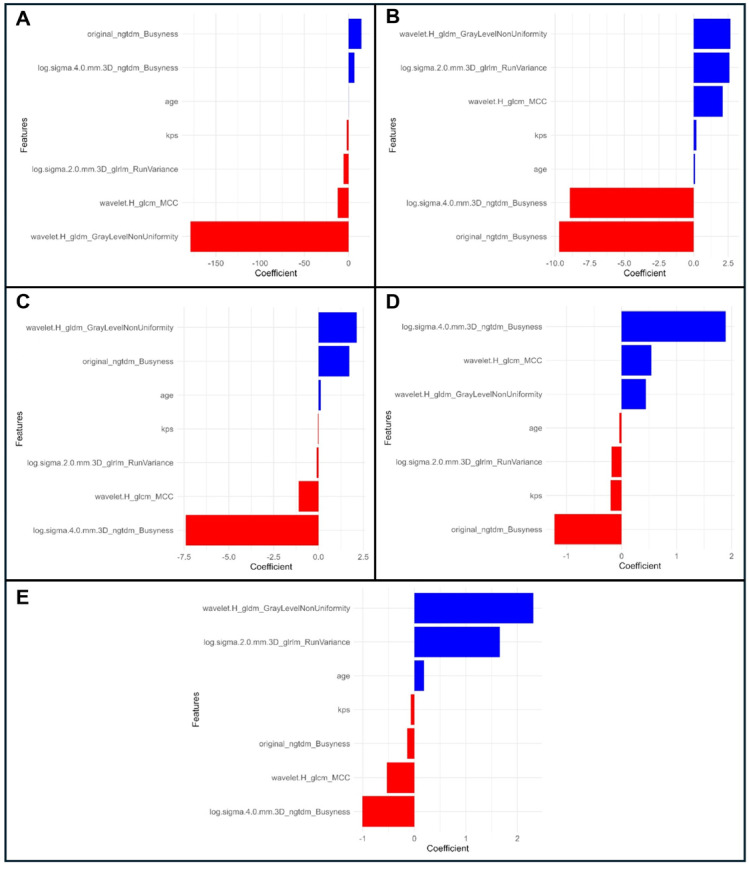
Representation of variable importance and their coefficients contributing to the combined model for the test datasets across folds 1 to 5, shown in panels (**A**–**E**).

**Figure 4 cancers-17-00280-f004:**
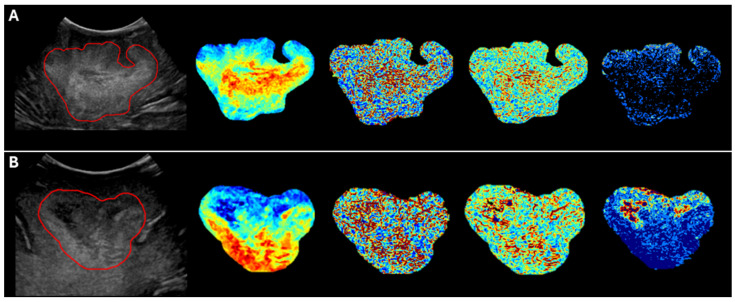
Representative examples of two patients: (**A**) a patient with short overall survival (104 days) and (**B**) a patient with long overall survival (947 days). From left to right: the original intraoperative ultrasound image with the tumor segmentation outlined in red, followed by texture maps of key radiomic features used in the predictive model. These include First-Order Median, Gray Level Run Length Matrix—Run Variance, Neighboring Gray Tone Difference Matrix—Busyness, and Neighboring Gray Tone Difference Matrix—Strength. The feature maps highlight patterns not visually discernible in the original ultrasound image, potentially providing valuable information for predicting overall survival.

**Table 1 cancers-17-00280-t001:** Patient demographics and clinical characteristics by center.

Center	n	OS	Age	KPS	Preoperative Tumor Volumen (cm^3^)	EOR Complete	EOR Incomplete
INCC	10	620.5 (375)	58.9 (19.8)	80 (20)	35.2 (30.5)	1 (10%)	9 (90%)
TMC	34	412 (811)	46.3 (10.9)	80 (10)	34.945 (29.095)	30 (88.2%)	4 (11.8%)
RHUH	57	335 (354)	62.6 (10.8)	80 (10)	31.66 (38.46)	16 (28.1%)	41 (71.9%)
CHRUT	13	392 (460)	57.6 (11.2)	80 (10)	35.2 (22.36)	12 (92.3%)	1 (7.7%)
Overall	114	382 (444.25)	56.9 (13.7)	80 (20)	32.685 (31.765)	59 (51.8%)	55 (48.2%)

OS = overall survival. KPS = Karnofsky performance status. EOR = extent of resection. INCC = Istituto Neurologico “Carlo Besta”. TMC = Tata Memorial Center. RHUH = Río Hortega University Hospital. CHRUT = Le Centre Hospitalier Régional Universitaire de Tours. Values are expressed as medians (IQR, interquartile range) or means ± SD (standard deviation), as appropriate.

**Table 2 cancers-17-00280-t002:** Distribution of variables between training and testing cohorts across folds.

**Fold**	Train OS	Test OS	*p*	Train Age	Test Age	*p*	Train KPS	Test KPS	*p*	Train Preoperative Tumor Volume	Test Preoperative Tumor Volume	*p*	Train EOR Complete	Test EOR Complete	*p*
1	385 (419.5)	328 (477.5)	0.89	57.79 (13.59)	53.26 (13.92)	0.17	80 (20)	80 (10)	0.11	33.69 (31.715)	26.7 (29.41)	0.33	44 (48.35%)	11 (47.83%)	0.32
2	379 (445)	385 (415.5)	0.89	57.51 (13.29)	54.39 (15.37)	0.38	80 (20)	80 (15)	0.3	32.39 (33.22)	38.49 (27.835)	0.82	45 (49.45%)	10 (43.48%)	0.46
3	379 (471)	436 (447)	0.9	55.77 (13.62)	61.26 (13.53)	0.09	80 (20)	80 (12)	0.99	30.99 (35.315)	33.11 (24.02)	0.77	39 (42.86%)	16 (69.57%)	0.15
4	385 (433.5)	371 (478)	0.72	56.55 (14.54)	58.17 (9.99)	0.53	80 (20)	80 (20)	0.68	32.39 (28.86)	35.2 (42.16)	0.75	46 (50.55%)	9 (39.13%)	0.21
5	381 (479)	383 (352)	0.95	56.77 (13.51)	57.32 (14.89)	0.88	80 (20)	80 (20)	0.87	32.685 (31.905)	32.215 (32.605)	0.47	46 (50%)	9 (40.91%)	0.19

OS = overall survival. KPS = Karnofsky performance status. EOR = Extent of resection. Values are expressed as medians (IQR, interquartile range) or means ± SD (standard deviation), as appropriate. The average performance of the radiomic, clinical, and combined Cox proportional hazards models across the five cross-validation folds was as follows: in the training cohort, the radiomic model achieved a mean C-index of 0.67 (95% CI: 0.59–0.75), in the clinical model, 0.70 (95% CI: 0.63–0.77), and in the combined model, 0.78 (95% CI: 0.72–0.84). In the testing cohort, the radiomic model had a mean C-index of 0.72 (95% CI: 0.57–0.86), in the clinical model, 0.73 (95% CI: 0.60–0.87), and in the combined model, 0.87 (95% CI: 0.76–0.98).

**Table 3 cancers-17-00280-t003:** Model performance metrics across folds for training and testing cohorts.

Fold	Model	Cohort	C-Index	Likelihood Ratio Test	Wald Test	Logrank Test
Fold 1	Radiomic	Train	0.65 (0.04)	27.00, *p* < 0.01	25.15, *p* < 0.01	38.20, *p* < 0.01
		Test	0.78 (0.05)	7.86, *p* = 0.35	6.09, *p* = 0.53	7.99, *p* = 0.33
	Clinical	Train	0.68 (0.03)	23.42, *p* < 0.01	21.25, *p* < 0.01	22.07, *p* < 0.01
		Test	0.82 (0.08)	13.56, *p* < 0.01	10.67, *p* < 0.01	13.28, *p* < 0.01
	Combined	Train	0.77 (0.03)	53.97, *p* < 0.01	46.24, *p* < 0.01	66.25, *p* < 0.01
		Test	0.91 (0.05)	31.36, *p* < 0.01	7.89, *p* = 0.72	21.17, *p* = 0.03
Fold 2	Radiomic	Train	0.67 (0.04)	25.29, *p* < 0.01	23.82, *p* < 0.01	34.17, *p* < 0.01
		Test	0.79 (0.06)	19.18, *p* < 0.01	14.05, *p* = 0.05	22.69, *p* < 0.01
	Clinical	Train	0.70 (0.04)	26.58, *p* < 0.01	24.33, *p* < 0.01	25.91, *p* < 0.01
		Test	0.68 (0.07)	7.17, *p* = 0.03	6.13, *p* = 0.05	6.34, *p* = 0.04
	Combined	Train	0.79 (0.03)	60.82, *p* < 0.01	52.27, *p* < 0.01	74.79, *p* < 0.01
		Test	0.91 (0.05)	42.84, *p* < 0.01	14.15, *p* = 0.44	33.25, *p* < 0.01
Fold 3	Radiomic	Train	0.70 (0.04)	32.01, *p* < 0.01	29.49, *p* < 0.01	35.26, *p* < 0.01
		Test	0.64 (0.08)	6.86, *p* = 0.44	4.10, *p* = 0.77	5.30, *p* = 0.62
	Clinical	Train	0.70 (0.04)	30.70, *p* < 0.01	28.04, *p* < 0.01	28.32, *p* < 0.01
		Test	0.71 (0.07)	4.99, *p* = 0.08	4.68, *p* = 0.10	5.01, *p* = 0.08
	Combined	Train	0.79 (0.03)	61.75, *p* < 0.01	53.30, *p* < 0.01	59.74, *p* < 0.01
		Test	0.77 (0.09)	9.30, *p* = 0.32	6.41, *p* = 0.60	8.18, *p* = 0.42
Fold 4	Radiomic	Train	0.64 (0.04)	17.61, *p* < 0.01	16.06, *p* = 0.01	10.71, *p* = 0.06
		Test	0.77 (0.08)	16.28, *p* = 0.01	10.61, *p* = 0.06	16.57, *p* = 0.01
	Clinical	Train	0.72 (0.04)	34.63, *p* < 0.01	30.22, *p* < 0.01	31.45, *p* < 0.01
		Test	0.66 (0.07)	3.69, *p* = 0.16	3.49, *p* = 0.18	3.66, *p* = 0.16
	Combined	Train	0.78 (0.03)	57.21, *p* < 0.01	45.94, *p* < 0.01	41.20, *p* < 0.01
		Test	0.89 (0.05)	28.45, *p* < 0.01	12.53, *p* = 0.25	27.89, *p* < 0.01
Fold 5	Radiomic	Train	0.69 (0.04)	18.77, *p* < 0.01	15.43, *p* = 0.02	23.50, *p* < 0.01
		Test	0.60 (0.10)	9.49, *p* = 0.15	4.50, *p* = 0.61	20.50, *p* < 0.01
	Clinical	Train	0.69 (0.03)	24.13, *p* < 0.01	23.19, *p* < 0.01	23.72, *p* < 0.01
		Test	0.79 (0.06)	15.00, *p* < 0.01	10.02, *p* < 0.01	11.41, *p* < 0.01
	Combined	Train	0.76 (0.03)	41.93, *p* < 0.01	40.57, *p* < 0.01	49.44, *p* < 0.01
		Test	0.87 (0.04)	23.21, *p* < 0.01	13.06, *p* = 0.11	19.42, *p* = 0.01

Concordance index (C-index) values are expressed as means ± standard deviation.

**Table 4 cancers-17-00280-t004:** Top Radiomic Features: Frequency, Coefficients, and Statistical Significance Across Folds.

Radiomic Feature/Image Filter	Folds Appeared	Average Coefficient	Predominant Direction	Average *p*-Value
NGTDM Busyness/LoG sigma 4.0	4 out of 5	−0.94	Negative	0.35
First-Order Median/Wavelet (H)	3 out of 5	0.02	Mixed	0.2
GLRLM Run Variance/LoG sigma 2.0	3 out of 5	0.52	Positive	0.03
GLCM MCC/LoG sigma 2.0	3 out of 5	−0.32	Negative	0.04
NGTDM Strength/LoG sigma 4.0	4 out of 5	−2.46	Negative	0.22
GLRLM Long Run High Gray Level Emphasis/Wavelet (H)	3 out of 5	0.61	Positive	0.08
GLSZM Small Area Emphasis/LoG sigma 3.0	2 out of 5	0.46	Positive	0.07
NGTDM Busyness/Original	4 out of 5	0.39	Positive	0.06

NGTDM = neighboring gray-tone difference matrix. GLRLM = gray-level run length matrix. GLCM = gray-level co-occurrence matrix. GLSZM = gray-level size zone matrix. MCC = maximal correlation coefficient. LoG = Laplacian of Gaussian.

## Data Availability

The raw data supporting the conclusions of this article will be made available by the authors on request.

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
