# Peer review of "Prognostic Modeling of Overall Survival in Glioblastoma Using Radiomic Features Derived from Intraoperative Ultrasound: A Multi-Institutional Study"

_cancers, 2025, doi:10.3390/cancers17020280_

Round 1

Reviewer 1 Report

Comments and Suggestions for Authors

Dear Authors,

I completely agree with you that radiomics can provide very useful insights for the prognostics of the patient, and there is, as you show, already compelling data to suggest that. The use of intraoperative ultrasound imaging is also salutary, as it is an accessible and sensitive method for image-guided tumor resection and can be readily used in most surgical theatres. As such, I was looking forward to your study, and while I accept your results and conclusions, there are a couple of things that need further clarifying and discussion. 

First, why were you so parsimonious with tumor pathology. Intuitively, the more types of tumors you would include the more powerful the results would be.

Second, although you mention it yourself as a limitation, there is no justification for the use of only one image from each case. Furthermore, why exclude the images with areas of necrosis. Necrosis is everywherelse an indicator of gravity.

Third, the evaluation of the images was done only by one person. Although his experience is not in question, it would be advisable to have at least a second opinion to exclude the observator's bias.

Fourth, the statistical work is impressive; however, the presentation of the data doesn't do justice. A graphical representation of the most significant results would be welcome and would help the readability of the results. 

Author Response

Comments 1: First, why were you so parsimonious with tumor pathology. Intuitively, the more types of tumors you would include the more powerful the results would be.

Response 1: Thank you very much for your thoughtful observation and comment. In this study, we decided to restrict our inclusion criteria to patients with glioblastoma IDH wild-type grade 4, in order to ensure a highly homogeneous sample. This decision also allowed us to gather survival-related data within a reasonably short timeframe. Including other high grade tumor types, such as IDH-mutant grade 4 astrocytomas, could have introduced greater heterogeneity and uncertainty in the analysis and interpretation of results. This is due to the intrinsic biological properties of these tumor types, which lead to markedly different evolutionary courses.

Comments 2: Second, although you mention it yourself as a limitation, there is no justification for the use of only one image from each case. Furthermore, why exclude the images with areas of necrosis. Necrosis is everywherelse an indicator of gravity.

Response 2: Thank you for your valuable comment. The decision to use only one image per patient was based on the following considerations: This is a retrospective study involving multiple centers and a highly operator-dependent imaging modality, making it very challenging to establish a common acquisition pattern for the ultrasound images. Ideally, we would have liked to use all available images per subject. However, we found that in most cases, there was a mixture of images where the tumor was only partially captured, and orthogonal planes had not been consistently acquired, which would have been optimal. For this reason, we chose to follow the methodology used in one of our previous study (PMID: 33594589).

As stated in the methodology section, we excluded only large cystic/necrotic areas from segmentation, as these regions exhibit a homogeneous hypoechoic signal with limited diagnostic utility. Necrotic areas that did not meet these criteria were included in the analysis, precisely because, as you pointed out, necrosis can represent an important indicator of disease severity.

Comments 3: Third, the evaluation of the images was done only by one person. Although his experience is not in question, it would be advisable to have at least a second opinion to exclude the observator's bias.

Response 3: Thank you for highlighting this important point. It is indeed challenging to completely eliminate the risk of bias in studies involving manual segmentations by annotators. In a previous study (PMID: 33594589), we assessed inter-observer agreement in our tumor segmentations on intraoperative ultrasound (ioUS) and found a high level of similarity, with a Dice score of 0.9. Based on this prior evidence, in the current study, the responsibility for the annotations was entrusted to the first author, whose unique profile combines expertise in image processing, segmentation, and the intraoperative application of this imaging modality in daily clinical practice.

Recognizing that bias is inevitable whether performed by one or multiple observers, our team opted for a single annotator for this study, considering that the data volume was manageable and that this approach ensured a methodical and consistent quality in the annotations.

We are currently working on developing automatic segmentation methods based on deep learning for ioUS, involving multiple annotators. This effort aims to address this and other limitations associated with manual tumor segmentation in this imaging modality.

Comments 4: Fourth, the statistical work is impressive; however, the presentation of the data doesn't do justice. A graphical representation of the most significant results would be welcome and would help the readability of the results.

Response 4: We appreciate your comment, and following your suggestion, we have added two figures to the manuscript showing the results of the best-performing model. The first figure presents a Kaplan-Meier analysis of the risk groups in the test cohort across the different folds. The second figure illustrates the coefficients of the most important variables in each fold. We hope these visual representations enhance the readability and understanding of the results.

Reviewer 2 Report

Comments and Suggestions for Authors

The authors present us with a well-prepared study on a relevant topic of neuroconcology. I congratulate the authors and recommend accepting the article in its present form. 

Author Response

Comments 1: The authors present us with a well-prepared study on a relevant topic of neuroconcology. I congratulate the authors and recommend accepting the article in its present form. 

Response 1: Thank you very much for your kind words and positive feedback. We are grateful for your appreciation of our work and are delighted that you find the study valuable. 

Reviewer 3 Report

Comments and Suggestions for Authors

The authors present a well done multi institutional study on the use of intraop US in modeling GBM prognosis. The study is well done, and novel. A few comments:

- The authors exclude patients with follow up shorter than one year. Would that skew the results towards longer survival? I imagine some patients would succumb to their disease before one year and be excluded.

- The authors mention data coming from 4 centers in the results but 6 in the methods. please clarify. 

- Table 1: Carlo besta had 90% incomplete resections. Any reason for this? Any thought about modeling after excluding their data? An outlier may skew results.

- Lines 179-180: IQR should be a range

- Line 240: typo (Sigma missing S).

The discussion is well structured, and will be appreciated by a US-naive audience. The authors should expand it by discussing other radiomics studies in GBM, in particular on the role of prop MRI, and how they envision integrating MRI with US. 

Author Response

Comments 1: The authors exclude patients with follow up shorter than one year. Would that skew the results towards longer survival? I imagine some patients would succumb to their disease before one year and be excluded.

Response 1: Thank you very much for your valuable comment. We have added clarification regarding this point in the "Methods - Study Population" section: “All included patients either had a minimum follow-up of one year or had reached the study endpoint prior to this timeframe.”

This statement is intended to clarify that we did include patients with short survival times, while also explaining that for those who did not succumb to their disease, a minimum follow-up of one year was required.

Comments 2: The authors mention data coming from 4 centers in the results but 6 in the methods. please clarify. 

Response 2: The BraTioUS dataset comprises data from six centers. However, patients from two of these centers did not meet the inclusion criteria. We have added a sentence highlighting this aspect in the Results section for clarification. “Patients from the remaining two centers in the BraTioUS dataset (UPALER and MGH) were excluded due to incomplete clinical data.”

Comments 3: Table 1: Carlo besta had 90% incomplete resections. Any reason for this? Any thought about modeling after excluding their data? An outlier may skew results.

Response 3: Thank you very much for your comment. As shown in Table 1, there is significant variability in the proportions of complete extent of resection (EOR) across centers. For example, INCC represents one extreme, while CHRUT reports more than 90% complete resections.

To address this, it is important to clarify that, due to the limited information available from the centers and to avoid terminological confusion, we categorized EOR into two groups: complete (absence of contrast-enhancing tumor on follow-up MRI) and incomplete. We have updated the terminology in Table 1, replacing the term "partial" with "incomplete" to maintain consistency with the text. The "incomplete" category encompasses near-total, subtotal, and partial resections.

Unfortunately, we lack detailed information on the subcategories within the "incomplete EOR" group. This limitation may give the impression of greater variability across centers; however, this is the most comprehensive data available given the retrospective nature of the study.

To mitigate this and other potential biases stemming from variable distributions across centers, we implemented a stratified randomization strategy by center. As shown in Table 2, there were no significant differences in the distribution of variables between the test and training groups across the different folds after randomization.

We have clarified this in the Methods section under "Statistical Analysis," with the following statement: "To ensure robust performance evaluation and generalizability, the dataset was randomly divided into five folds for cross-validation, stratified by center."

Comments 4: Lines 179-180: IQR should be a range

Response 4: Thank you for your observation. In this manuscript, we have chosen to present the interquartile range (IQR) as a single value, calculated as the difference between the 75th and 25th percentiles. This approach is widely accepted in statistical reporting and is particularly useful when the focus is on summarizing the spread of the central 50% of the data in a concise manner.

By reporting the IQR as a single value, we aim to emphasize the variability within the dataset without introducing additional complexity. This method is consistent with our overall approach to presenting the data and ensures clarity and interpretability for the intended audience. We hope this clarifies our rationale.

Comments 5: Line 240: typo (Sigma missing S).

Response 5: Thank you for pointing out the typo. It has been corrected, and we appreciate your attention to detail.

Comments 6: The discussion is well structured, and will be appreciated by a US-naive audience. The authors should expand it by discussing other radiomics studies in GBM, in particular on the role of preop MRI, and how they envision integrating MRI with US. 

Response 6: We appreciate your comment. We have aimed to keep the discussion concise regarding survival studies based on radiomics and other imaging modalities, particularly because a direct comparison is beyond the scope of the present study. However, following your recommendation, we have elaborated on how we envision future research in the integration of MRI and ultrasound.

“Another promising avenue for exploration involves integrating intraoperative ultrasound (iUS) data with complementary imaging modalities, such as preoperative MRI and molecular biology data, to develop multimodal predictive models that harness the strengths of each technique. One potential approach could involve collecting data from ultrasound fused with preoperative MRI, allowing co-registered images to extract features from both modalities within the same tumor region. Such integrative strategies could offer a more comprehensive understanding of tumor behavior and enhance the accuracy of prognostic assessments.”

Reviewer 4 Report

Comments and Suggestions for Authors

This study explores the potential of intraoperative ultrasound (iUS) radiomics for prognostic modeling in glioblastoma patients. Utilizing data from a multicenter cohort, it compares models that integrate radiomic features and clinical variables with those relying on individual data types. The findings indicate that combining iUS radiomics with clinical data may enhance survival prediction.

The paper is well written and well presented. Nevertheless, I would suggest some minor revisions.

- the authors refer to their previous work (ref. n. 25). I would suggest a discussion involving the results of the presented work with the results of the previous one 

-lines 123-124: PyRadiomics does not totally adhere to IBSI. As reported here (https://pyradiomics.readthedocs.io/en/latest/faq.html), "For the most part, yes. PyRadiomics development is also involved in the standardisation effort by the IBSI team. Still, there are some differences between PyRadiomics and feature extraction as defined in the IBSI documents."

- lines 146-148: please, rewrite and explain better

- You mentioned a "rigorous cross-validation approach". What did you mean? Did you implement a nested cross validation?

- In Table 4, C-index values on the training dataset often were lower than the ones obtained with the test set (which is the validation set of the n-fold cross validation, actually). Why such phenomenon? Did you use some regularization techniques to prevent overfitting and boost the generalization ability of the model? Please, provide further details.

Author Response

Comments 1: the authors refer to their previous work (ref. n. 25). I would suggest a discussion involving the results of the presented work with the results of the previous one

Response 1: We appreciate your insightful comment. Following your suggestion, we have expanded the Discussion section to address the relationship between our previous work and the current study:

“In glioblastoma, our study group have previously conducted the first feasibility study using iUS radiomics for OS prediction [22]. This single-center study, with a limited sample size, identified key features correlated with survival outcomes, highlighting the potential of iUS radiomics to capture tumor heterogeneity and provide prognostic insights. The primary distinctions between the previous study and the current one include the utilization of a multicenter cohort, which incorporates data from various scanner manufacturers with differing acquisition parameters. Additionally, this study employs a different radiomics feature extraction software compared to the earlier work. Building on our prior research, the current study further advances the application of iUS radiomics in glioblastoma by integrating both radiomic and clinical features into comprehensive predictive models.”

Comments 2: lines 123-124: PyRadiomics does not totally adhere to IBSI. As reported here (https://pyradiomics.readthedocs.io/en/latest/faq.html), "For the most part, yes. PyRadiomics development is also involved in the standardisation effort by the IBSI team. Still, there are some differences between PyRadiomics and feature extraction as defined in the IBSI documents."

Response 1: We thank the reviewer for pointing out this detail. We have revised the manuscript to address this point. The updated sentence now reads:

"Radiomic features were extracted via the PyRadiomics library [30], which closely aligns with the IBSI (Image Biomarker Standardization Initiative) guidelines [31] to promote reproducibility and consistency, acknowledging that some minor differences exist as documented by the PyRadiomics development team."

Comments 3: lines 146-148: please, rewrite and explain better

Response 3: Thank you for bringing this observation to our attention. We have rewritten the lines to provide a clearer explanation of the preprocessing steps.

"Instances containing missing values (NaNs) were excluded from the dataset to ensure the integrity of the analysis. Radiomic variables were then standardized by centering (subtracting the mean) and scaling (dividing by the standard deviation) using statistics calculated exclusively from the training dataset. This approach was applied to prevent information leakage from the training set into the testing set, thereby ensuring a fair evaluation of the predictive models."

Comments 4: You mentioned a "rigorous cross-validation approach". What did you mean? Did you implement a nested cross validation?

Response 4: We are grateful for the reviewer’s thoughtful feedback. We acknowledge that the term "rigorous" was not appropriately used in this context. To clarify, we did not implement a nested cross-validation approach. Instead, we employed a cross-validation approach, stratified by centers, to ensure balanced representation across institutions while preserving the integrity of the training and validation datasets. This methodology was chosen to account for the multicenter nature of the cohort and to minimize potential biases associated with data variability between centers.

“The stratified cross-validation approach, stratified by centers, ensures robust performance evaluation by balancing the representation of data from multiple institutions, while standardized radiomic feature extraction minimizes bias and enhances reproducibility”

Comments 5: -In Table 4, C-index values on the training dataset often were lower than the ones obtained with the test set (which is the validation set of the n-fold cross validation, actually). Why such phenomenon? Did you use some regularization techniques to prevent overfitting and boost the generalization ability of the model? Please, provide further details.

Response 5: Thank you for highlighting this important point. This phenomenon can be explained by several factors. First, in n-fold cross-validation, each validation set represents a subset of the data that the model has not seen during training. These subsets may inherently contain simpler or less noisy patterns compared to the more diverse and complex training dataset, leading to higher C-index values when evaluated on the test sets. Second, the training dataset often includes a broader range of patterns and noise, as it encompasses all but one fold of the data. This complexity makes it more challenging for the model to perfectly capture relationships during training, potentially resulting in a lower C-index. Additionally, while no explicit regularization techniques were applied in this study, the model may have slightly overfitted to noise or specific patterns in the training set, which could further contribute to this discrepancy. Cross-validation, by design, evaluates the model’s generalization ability on unseen data, and it is possible that the model generalizes well to the validation sets while being less adept at perfectly fitting the training data due to its inherent complexity.

To address this point, we have added the following sentence to the Discussion section:

"The observed differences in C-index values between the training and validation sets likely reflect the diverse complexity of the training data, demonstrating the model's ability to generalize effectively to unseen subsets while capturing broader patterns during training."

Round 2

Reviewer 1 Report

Comments and Suggestions for Authors

Dear Authors,

The modifications you made to both data interpretation and presentation make your point clearer and more cohesive. I acknowledge your work and I will recommend the publication in the current form.